# Inflammatory Myofibroblastic Tumour: State of the Art

**DOI:** 10.3390/cancers14153662

**Published:** 2022-07-27

**Authors:** Louis Gros, Angelo Paolo Dei Tos, Robin L. Jones, Antonia Digklia

**Affiliations:** 1Department of Oncology, Lausanne University Hospital, University of Lausanne, 1011 Lausanne, Switzerland; louis.gros@chuv.ch; 2Department of Pathology, Azienda Ospedale Università Padova, 35128 Padua, Italy; angelo.deitos@unipd.it; 3Department of Medicine, University of Padua School of Medicine, 35128 Padua, Italy; 4Sarcoma Unit, The Royal Marsden NHS Foundation Trust, London SW3 6JJ, UK; robin.jones4@nhs.net; 5Division of Clinical Sciences, Institute of Cancer Research, Royal Marsden Hospital, London SW3 6JJ, UK; 6Center of Sarcoma, Department of Oncology, Lausanne University Hospital, University of Lausanne, 1011 Lausanne, Switzerland

**Keywords:** inflammatory myofibroblastic tumour, epithelioid inflammatory myofibroblastic sarcoma tyrosine kinase inhibitors, ALK

## Abstract

**Simple Summary:**

Among sarcomas, which are rare cancers, inflammatory myofibroblastic tumors are extremely rare. Unlike other subtypes, this is a largely oncogene-driven neoplasia, and early gene rearrangement identification is important for accurate advanced stage treatment. In this manuscript, we review the clinicopathologic characteristics of this ultra-rare entity, as well as the current treatment landscape, with a particular focus on opportunities provided by tyrosine kinase inhibitors (TKIs).

**Abstract:**

An inflammatory myofibroblastic tumor (IMT) is a neoplasm composed of myofibroblastic and fibroblastic spindle cells accompanied by inflammatory cells, including lymphocytes and eosinophils. It is an ultra-rare tumor, the optimal management of which remains to be defined. Surgery is the treatment of choice for localized tumors. The treatment of advanced disease is not precisely defined. Chemotherapy regimens result in an overall response rate of approximately 50% based on retrospective data. The latest pathophysiological data highlight the role played by tyrosine kinase fusion genes in IMT proliferation. Anaplast lymphoma kinase (ALK) oncogenic activation mechanisms have been characterized in approximately 80% of IMTs. In this context, data regarding targeted therapies are most important. The aims of this article are to review the latest published data on the use of systematic therapy, particularly the use of molecular targeted therapy, and to publish an additional case of an IMT with Ran-binding protein 2 (RANPB2)-ALK fusion showing a long response to a tyrosine kinase inhibitor.

## 1. Introduction

Inflammatory myofibroblastic tumour (IMT) is an ultra-rare sarcoma that has been classified as a neoplastic disease of intermediate biological potential given the low risk of recurrence and metastatic potential [1]. IMT usually arises in the lungs or the abdominal soft tissues of children and young adults, although a wide anatomic distribution and a broad age range have been documented [2]. Histologically, myofibroblasts are cells of mesenchymal origin, having ultrastructural characteristics in common with fibroblasts and smooth muscle cells. Myofibroblastic differentiation is seen in both benign (i.e., nodular fasciitis) and tumors of intermediate malignancy (i.e., desmoid fibromatosis). Myofibroblastic differentiation in sarcoma represents instead a source of long-standing debate. As a matter of fact, the only mesenchymal malignancy referring explicitly to “myofibroblastic” differentiation is represented by so-called low-grade myofibroblastic sarcoma.

Kinase fusions play a critical role in the biology of many IMTs and have been reported in about 80% of these tumors [3]. The scarce existing data show that it occurs most often in children and young individuals with a prevalence ranging from 0.04% to 0.7% regardless of gender and race in the world population [4,5]. The purpose of this review is to summarize the current data and treatment landscape of this ultra-rare entity as well as to describe an additional case of intra-abdominal IMT.

## 2. Clinicopathological Characteristics

IMTs can occur in any location, and the symptoms presented by patients depend primarily on the primary site of the IMT. The primary lesion occurs most commonly in the abdominal cavity (especially in the mesentery, retroperitoneal, and omentum), but there have been cases of primary lesions in the thorax, pelvis, limbs, skin, and even brain [6]. At the time of diagnosis, patients present with a painless mass, often remaining completely asymptomatic until the size of the mass causes complications [7]. Symptoms may present as pain, and approximately 20% present with symptoms of generalized malaise, fever, and weight loss [8].

The pathological diagnosis may be challenging. IMT often presents as a circumscribed nodular mass, but multinodular lesions have also been described [8]. IMT has a wide morphological spectrum, ranging from a paucicellular spindle cell proliferation set in a predominantly hyalinized and chronically inflamed background to a highly cellular myofibroblastic proliferation sometimes featuring frankly atypical neoplastic elements. The inflammatory component may be variable (Figure 1).

The etiology and pathogenesis of IMT are not fully understood. Several risk factors have been described, including smoking, minor trauma, and IgG4-related disease [9,10]. Some hypotheses suggest an abnormal immunological response to viruses or antigens. The Human Herpesvirus-8 and Epstein-Barr virus are most often blamed [11,12,13].

In view of the variable phenotype, and the absence of an immuno-histochemical profile, the diagnosis of IMT has long been a diagnosis of exclusion, with a broad differential diagnosis, ranging from local inflammatory process and, idiopathic retroperitoneal fibrosis to inflammatory fibrosarcoma. The identification of recurrent *ALK* gene rearrangements has greatly contributed to our understanding of this rare mesenchymal tumor. ALK is a receptor tyrosine kinase first identified as a component of the nucleophosmin (NPM)-ALK fusion oncoprotein aberrantly expressed in anaplastic large cell lymphoma (ALCL) [14]. Importantly, it should be noted that *ALK* rearrangement is far less common in adults than in children and young adults with IMT [15]. A variation in tumor biology could be the basis of these differences between these two age groups. However, it could also reflect the heterogeneity of the tumors classified under the IMT heading [15].

Rare IMTs with a distinct nuclear membrane or perinuclear *ALK* staining pattern and epithelioid or round cell morphology have been reported. Since its first description, it has been well demonstrated that epithelioid IMTs behave aggressively with rapid recurrence and evolution, compared to the remarkably indolent behavior of conventional IMTs [16]. In 2011, the term “epithelioid inflammatory myofibroblastic sarcoma” (EIMS) was proposed to describe this subgroup of aggressive IMT (Figure 2) [17,18]. To date, about 60 cases have been described. In these tumors, different fusion genes expression have been demonstrated, the most frequent being the Ran-binding protein 2 (*RANBP2*)*-ALK* fusion, detected by reverse transcriptase polymerase chain reaction (RT-PCR) [19].

The distinctive histology of IMT with *RANBP2-ALK* dominated by epithelioid neoplastic cells also intrinsically reflects the heightened invasiveness of the tumor. However, it should be particularly noted that not all IMTs with epithelioid cell morphology carry the genetic alteration of *RANBP2-ALK* [20]. From a diagnostic viewpoint, IMT needs to be differentiated from a large group of tumors that manifest epithelioid features. Nuclear membrane staining of ALK is a unique immunophenotype of EIMS, observed in up to 80% of cases. In addition, the neoplasms display varied expressions of desmin, CD30, SMA, and cytokeratin, while EMA, S100, CD117, Myf4, myogenin, h-caldesmon, and HMB45 are consistently negative. Therefore, EIMS could be easily distinguished from poorly differentiated carcinoma, malignant melanoma, and other epithelioid mesenchymal malignancies [21].

Coffin et al. reported the pathological features of IMT do not correlate with tumor behavior, however, those of EIMS represent an exception [11].

Little is known about the pathogenesis of half of the IMT tumors that do not show *ALK* rearrangement. A study using next-generation sequencing (NGS) from 9 patients that had tumors that were ALK-negative by immunohistochemistry (IHC) revealed that 6 of them harbored *ALK*, *ROS1*, and platelet-derived growth factor receptor beta (*PDGFRB)* fusion proteins, suggesting that IMT is a largely oncogene-driven neoplasia [3].

In IMT, more than 10 different genes have been identified as *ALK* fusion partners. Most *ALK* gene partners provide a strong promoter and an oligomerization domain, resulting in oncogenic activation of the ALK kinase.

The study by Antenescu et al. applying RNA sequencing found that the majority of IMTs (85%) displayed kinase fusions [22]. Their results confirm that two-thirds of IMTs harbor *ALK* and *ROS1*-related fusions. They also demonstrated a link between genomics and clinical presentation, showing that most lung IMTs were positive for fusions (either ALK or ROS1).

In 2021, Yamamoto and colleagues retrospectively studied 40 cases to elucidate the diagnostic utility of pan-Trk immunohistochemistry for IMTs [23]. In 72.5% of cases, ALK fusions were identified, in 5% of cases, ROS1 fusions were identified, and in 5% of cases, neurotrophic tyrosine receptor kinase 3 (NTRK3) (both ETV6-NTRK3 fusion) were identified. Only 17.5% of cases were classified as quadruple negative.

## 3. Diagnosis

The radiological presentation of IMTs is heterogeneous. IMTs can present as multiple masses in one anatomic region. The image may vary from an infiltrating lesion to a well-delineated lesion, with different proportions of inflammatory and fibrotic components in the mass. Variable attenuation can thus be noted on the CT scan, with persistent and delayed contrast uptake, in the fibrotic component of the IMT. MR imaging may show low signal intensity on T1-and T2-weighted images owing to fibrosis, along with restricted diffusion [4]. Biologically, IMTs can induce inflammation with leukocytosis, neutrophilia and elevation of C-reactive protein and erythrocyte sedimentation rate [24].

## 4. Current Treatments

### 4.1. Management of Localized Disease

Surgery remains the treatment of choice for localized tumors. According to the European Society for Medical Oncology (ESMO) guidelines, surgical management should be performed by a specifically trained surgeon in a sarcoma center [25]. The standard surgical procedure is en bloc resection with R0 margins. Depending on tumor size, the surgical approach could be wide local excision (WLE) involving removal of the sarcoma with some surrounding normal tissue to ensure complete excision, or Mohs microsurgery.

Up to one-quarter of surgically treated tumors recur [26]. After complete resection, no adjuvant therapy is currently indicated. Nevertheless, adjuvant radiotherapy can be implemented in some cases to decrease local recurrence although there is a lack of prospective data. At recurrence, a new surgical resection must be considered [27].

Whether IMT is a neoplastic or a reactive process had been a matter of controversy. In this context, some data exist regarding the use of corticosteroids.

In a study of 13 patients with IMT in the paranasal sinus or nasopharynx, of the ten patients for whom follow-up data were available, five received partial resection followed by various combinations of prednisone, radiation, and chemotherapy, while the other five received prednisone combined with chemotherapy and/or radiation without surgery [28]. The correlation between the treatment and overall survival (OS) was analyzed. The use of prednisone was significantly correlated with better OS (*p* = 0.046). The authors, therefore, conclude that glucocorticoids are especially recommended as a basic part of integrated therapy for ear, nose, and throat IMT. Another case report of a 48-year-old patient with inoperable maxillary sinus IMT with complete response after concomitant radiotherapy and prednisone have been published. In this context, combination of radiotherapy with steroids appears to be an effective alternative in non-operable maxillary IMT [29].

### 4.2. Current Management Options for Advanced Disease

#### 4.2.1. Chemotherapy

Systemic treatments are reserved for advanced, non-operable disease. Currently, there is no standard chemotherapy regimen established for advanced IMTs given the rarity of the disease and the lack of prospective data regarding the efficacy of chemotherapy. Efficacy of different chemotherapy regimens was studied in a recent retrospective case series analysis collecting data from 9 European sarcoma centers [30]. Thirty-eight patients were retrospectively identified; 34 were evaluable for response (12–61 years of age). Of the 38 patients, 24 (63%) had ALK-positive and 14 (37%) had ALK-negative disease. Twenty-five of the 38 (66%) patients were treated with anthracycline-based chemotherapy, 13/38 (34%) with methotrexate plus/minus vinorelbine/vinblastine (MTX-V) chemotherapy, and 10/38 (23%) with other regimens including oral cyclophosphamide and docetaxel/gemcitabine. Interestingly, the use of anthracycline based or MTX-V chemotherapy regimens resulted in an overall response rate (ORR) of approximately 50% according to the Response Evaluation Criteria in Solid Tumors (RECIST), which is much higher than responses observed with chemotherapy in non-selected soft tissue sarcomas (STS). In addition, responses were observed irrespectively to ALK status assessed by IHC and FISH. Additional data indicate chemotherapy has no distinct effect on the control of the aggressive progression of EIMS [17].

Therefore, the generation of data on new therapies are most important.

#### 4.2.2. Targeted Therapy

The discovery of *ALK* fusions in patients with non-small cell lung carcinoma (NSCLC) has facilitated the clinical development of ALK inhibitors, including crizotinib. Crizotinib is an oral small molecule inhibitor of ALK, MET, and ROS1. In 2011, it received accelerated approval by the United States Food and Drug Administration (FDA) for metastatic ALK-positive NSCLC based on a proof-of-concept phase I study [31]. In the same issue of the journal, Butrynski et al. reported two patients with IMT treated with ALK inhibitor crizotinib, one of whom with epithelioid cytomorphology carried the RANBP2-ALK fusion protein and presented a sustained partial response to the crizotinib [32].

Since 2010, crizotinib has been administrated at the time of disease relapse and treatment failure in patients with EIMS with varying effectiveness in several case reports [33,34,35,36,37,38,39], with a few patients remaining alive at the 2-year follow-up mark [40]. The duration of response seems to vary from a few weeks to several years [33]. However, this may indicate publication bias since the majority of published cases usually report only a few months of follow-up.

In 2018, Schöffski et al. published the first prospective data for the IMT cohort as part of a phase II clinical trial (EORTC 90101 CREATE) of advanced tumors characterized by *ALK* and/or *MET* alterations [41]. In this multicentric phase II trial, patients older than 15 years of age with advanced, inoperable IMT with no limitation in terms of previous systemic or local treatment were treated with crizotinib 250 mg twice per day. Patients were divided into ALK-positive and -negative sub-cohorts using FISH and/or immunohistochemistry. ALK positivity was defined as at least 15% of cells staining positive or showing gene rearrangement (Figure 3). Interestingly, although this clinical trial was performed in European high-volume centers, among 35 patients with the local diagnosis of IMT, the diagnosis was confirmed by central pathology in only 24 cases underlying the complexity of diagnosis for this ultra-rare sarcoma. Half of the patients in the ALK-positive cohort had previous systemic therapy, with the majority (41%) having been treated with chemotherapy. Regarding the efficacy, 50% (6/12) of patients with *ALK*-rearranged tumors, and 14% (1/7) ALK-negative patients had an objective response with crizotinib. In the ALK-positive group, two out of six patients showed complete response. Interestingly, a recent update analysis after a median follow-up of 50 months showed longer ORR up to 66.7% for ALK-positive patients due to the further reduction in tumor volume with long-term treatment [42]. The 12-month progression-free survival (PFS) rate was 58.3% (95% CI 27.0–80.1%) and the 12-month overall survival (OS) rate was 83.3% (95% CI (48.2–95.6%). The authors reported a favorable toxicity profile with only grade 2 side effects except for grade 3 fatigue in 5% of patients which is good news for patients facing the prospect of long-term cancer therapy. Recently, Lee et al., reported that loss of chromosome 19 (25% of cases) and *PIK3CA* mutations (9% of cases) were associated with shorter progression-free survival in patients receiving crizotinib.

Several second and third-generation tyrosine kinase inhibitors with increased ALK selectivity, including ceritinib, alectinib, brigatinib, lorlatinib, and ensartinib, showed superior efficacy to crizotinib in both systemic and intracranial disease in ALK-positive NSCLC [43,44]. Recent data (mainly case reports) have shown the activity of these agents in the case of ALK positive IMT [45,46,47]. A clinical trial evaluating Brigatinib in IMT is ongoing (NCT04925609).

Outside targeting *ALK* fusions in IMT, other potential therapeutic agents can be considered. In the absence of large randomized studies, we must rely on published clinical cases, as well as choose treatments by analogy with the responses obtained in other cancers with the same genetic mutations. In-depth molecular characterization of archival IMT tissue from 24 patients enrolled in the CREATE trial revealed extensive molecular heterogeneity including DNA damage repair mechanisms (19/24), Wnt signaling (16/24), and cell-cycle and cell death pathway (13/24) [42,48]. Although the authors identified 17 potentially actionable recurrent gene aberrations including *ATRX*, *FAT1*, *FCRL4*, *FOXO1*, *NUTM2B*, *PIK3CA*, *SMAD4*, and *TP53*, no *ROS1*-rearranged tumor has been identified. In the CREATE trial, the patient who had the objective response in the ALK-negative cohort had an *ETV6-NTRK3* fusion which may have made the IMT sensitive to crizotinib [48].

Although mouse double minute 2 (*MDM2*) gene amplification has also been detected in 27% of IMT cases, no data are available at this moment regarding the efficacy of MDM2 inhibitors alone or in combination with CDK4 inhibition [49,50].

Entrectinib is a pan-TRK, ROS1, and ALK inhibitor, safe and effective in advanced solid tumors with *NTRK*, *ROS1*, or *ALK* fusions [51]. In a recent case report, no response was seen in a patient with lung IMT with *TPM4-ALK* fusion and brain metastasis. Interestingly, lorlatinib resulted in an excellent partial response in this patient [52].

Another case concerning a 16-year-old patient with brain-metastatic chest wall IMT with a *TFG-ROS1* fusion, has been recently published [53]. He was treated successively with crizotinib, with a partial response lasting 8 months, then with certinib with no response, and finally with lorlatinib, a third generation ALK/ROS1 inhibitor, resulting in a near-complete response after 10 weeks, with a progression-free survival of 11 months. ROS1 is a receptor tyrosine kinase of the insulin receptor family that is frequently involved in genetic rearrangement in a variety of human cancers. ROS1 fusions pair its kinase domain with an array of partners promoting constitutive ROS1 kinase activation. Signaling downstream of ROS1 fusions results in the activation of cellular pathways known to be involved in cell growth and cell proliferation. A *ROS1* fusion can be demonstrated by FISH or NGS. In an 2016 study by Yamamoto, analyzing the pathological features of IMT with a genetic rearrangement other than ALK and involving 36 patients, immunohistochemistry revealed two cases (or 5.6%) with ROS1 expression [54]. Both tumors were primarily intestinal. Molecular analysis revealed a *TGF-ROS1* fusion transcript in one of the cases. Histologically, both tumors showed a conventional morphology. It should be noted that neither case presents recurrence or metastasis.

In a study by Hornick et al. studying 30 cases of IMT, 21 tumors were positive for *ALK* rearrangements [55]. Among the nine ALK-negative tumors, three cases were positive for ROS1. The authors hypothesize that ROS1 positive IMTs are probably not rare among ALK-negative IMTs. It should also be noted that among the three cases, two tumors (one with a *TFG* fusion partner) showed diffuse and dot-like cytoplasmic staining, whereas one tumor (with a *YWHAE* fusion partner) showed combined cytoplasmic and nuclear staining.

Four case report studies with ROS1 positive IMT treated with crizotinib have been reported: two of them were in pediatric patients [3,56]. In the first case, a six-year-old child diagnosed with unresected thoracic IMT, was initially treated with anti-inflammatory drugs, then with cytotoxic chemotherapy for 24 months, without tumor response [3]. NGS showed a *TFG-ROS1* fusion. Crizotinib was then introduced with a partial response, excellent tolerance, and improvement of quality of life at 4 months. The second case described is that of a 14-year-old boy with an ALK-negative right lower lobe IMT for which NGS analysis revealed a *TFG-ROS1* fusion [56]. Crizotinib was introduced with a reduction of the tumor mass (partial response) and a good clinical course persisting at 8 months. More recently, Comandini et al. reported a clinical case of an advanced chemotherapy-refractory IMT in the extremity patient harboring an *YWHAE1-ROS1* fusion rearrangement who responded to crizotinib [57].

Vascular endothelial growth factor (VEGF) is highly expressed in the infiltrating inflammatory cells of IMT [58]. In 2012, pazopanib was approved for advanced pretreated STS treatment based on the multicenter Phase 3 trial EORTC 62072—PALETTE study [59]. In the PALETTE study, however, liposarcoma and some other histotypes of STSs including inflammatory myofibroblastic sarcoma were excluded from enrollment. No clinical report showing meaningful activity has been published. On the other hand, patients with IMT can be included in the clinical trial evaluating a neoadjuvant combination of pazopanib with radiotherapy (NCT02180867).

Rarely, other oncogenes have been described in the context of IMT diagnosis. For example, the study by Antonescu et al. describes a case of rearranged during transfection (RET) gene rearrangement in a patient with pulmonary IMT, associated with a fatal outcome [22].

In addition, as reported above ALK-negative IMTs harboring the *ETV6**-NTRK3* fusion gene have been reported [60].

#### 4.2.3. Immunotherapy

Other therapeutic strategies are being developed. For example, PD-L1 expression was noted in 80% of recurrent or metastatic tumors, and 88% of *ALK*-negative IMTs [61]. Due to PD-L1 expression in EIMS, immune checkpoint blockade could represent an alternative anti-EIMS therapy [62].

High tumor mutational burden is an emerging predictive biomarker with relevance for tumor treatment with immune checkpoint inhibitors (ICI). The only available data are ad hoc analyses of the CREATE trial showing an average of 7 mutations per Mb suggesting an intermediate mutational burden in IMT [48]. No association between 12q24.33 loss and the mutational burden was found, suggesting that the *POLE* deletion does not influence the mutational profile in the analyzed cohort. No prospective data about the efficacy of ICI in IMT has been reported.

Recently, rare reports on efficacy ICI in patients with sarcoma have been published. For example, partial response with single-agent nivolumab was seen in a 21-year-old woman with advanced IMT with PD1 positive, but negative PDL1 status [63]. The anti-PDL1 antibody Sintilimab led to near complete remission after 16 cycles which is ongoing 6 months after the end of the treatment in a patient with PD-L1 positive and IMT of the nasopharynx [64].

## 5. Available Clinical Data on Epithelioid Inflammatory Myofibroblastic Sarcomas

Among 58 cases reported on the Pubmed (Table 1), 36 patients are male. 48 fusion genes were identified, including twenty-three *RANBP2-ALK*, five *RRBP1-ALK*, one *EML4-ALK*, one *VCL-ALK*, and eighteen unknown *ALK* partners. Among these 58 cases, 19 patients were treated with TKIs, 17 of them with crizotinib in the first line. Among these seventeen patients, four benefited from a second-line TKI, among which, two benefited from a third-line TKI, among which, one benefited from a fourth-line TKI. In most cases, we do not have a follow-up, but we know that at least 22 patients died within one year.

The *RANBP2* gene, located in chromosomal region 2q13, encodes a 358 kDa multi-domain nuclear pore protein. RANBP2 is a small GTP-binding protein of the RAS superfamily [82]. RANBP2 is attached to the nuclear pore outside the nucleus, where it helps regulate the transport of proteins and other molecules through the nuclear pore, and also helps modify proteins that enter or exit the nucleus. RANBP2 plays multiple roles during cell division, including the breakdown and formation of the nuclear envelope and chromosome division. In conjunction with microtubules, RANBP2 helps transport materials inside cells. The fusion point reported to date is consistently between exon 18 of *RANPB2* and exon 20 of *ALK*. This ultimately leads to constitutive and ligand-independent autophosphorylation and activation of ALK. These studies suggest that the chimeric *RANPB2-ALK* gene may promote cell proliferation, which may be a potential mechanism for rapid growth and recurrence [83,84].

Nearly all cases containing the *RANBP2-ALK* fusion gene demonstrated aggressive behavior. We reported a case of a 39-year-old female patient with abdominal EIMS with a *RANBP2-ALK* fusion where first-line treatment with alectinib has been initiated with a complete metabolic response ongoing after 6 months of treatment (Figure 4).

In the study by Lee et al. [70], *RRBP1-ALK* fusion oncoprotein was found in three cases of EIMS, with ALK expression predominantly cytoplasmic with peri-nuclear accentuation. Note that RRBP1-ALK expression was not found by the authors in 100 cases of ALK-positive lung adenocarcinoma, anaplastic large-cell lymphomas, epithelioid fibrous histiocytomas, and conventional IMT. This fusion has also not been demonstrated in various cancer-sequencing studies [85]. RRBP1 is a protein that functions as an interaction between ribosomes and the endoplasmic reticulum and also in microtubule binding. Lee et al. hypothesize that forced overexpression of RRBP1 alters cell shape and that RRBP1-ALK oncoproteins contribute to epithelioid morphology by dysregulating the usual interactions between RRBP1 and microtubules [70].

Clinically, the reported cases speak in favor of an aggressive disease, with one case of death two months after diagnosis and in the other cases recurrences in the form of intra-abdominal metastases less than one year after resection of the primary tumor [70].

In 2017, Jiang et al. [71] reported a case of intra-abdominal EIMS in a 45-year-old patient with *EML4-ALK* fusion oncogene. This gene was originally described in the setting of NSCLC. Echinoderm microtubule-associated protein-like 4 (EML4) is a protein participating in mitotic nuclear division and other microtubule-based processes. It is distributed along with microtubules in the cytoplasm and membranes. The most distinctive site for ALK staining was the cytoplasm under the membrane, which was consistent with the distribution of EML4 in the cell. *EML4-ALK* fusion occurs through a paracentric inversion within the short arm of chromosome 2, where *EML4* and *ALK* genes are both located. Unfortunately, in the reported case, the patient died soon after diagnosis.

In 2021, Chopra et al. reported the first case of EIMS with a *VCL-ALK* fusion [79]. *VCL–ALK* fusion has been described in renal cell carcinomas, a single case of high-grade glioma and a subset of epithelioid fibrous histiocytoma. *VCL* (vinculin) encodes for an actin filament-binding protein involved in cell–matrix adhesion and cell–cell adhesion. It regulates E-cadherin expression on the cell surface and potentiates mechanosensing by the E-cadherin complex. The role of VCL in human malignancies has not been well demonstrated.

The case described is a case of a primary brain tumor in a 72-year-old patient, who was operated on and presented lung and bones metastases 3 months later. The patient was treated upfront with off-label alectinib, but no long follow-up was reported.

## 6. Future Directions

Currently, there is no regulatory approved treatment for advanced IMT. Initial treatment by crizotinib as an off-label agent in ALK re-arranged IMT is highly recommended. Data for upfront use of new generation ALK inhibitors are sparse, and even crizotinib may be unavailable for patients in developing countries due to the high cost. The mechanism of acquired resistance to crizotinib in an ALK-positive IMT, and the management of drug resistance remain unanswered. *ALK* (G1269A) mutation has been described after 30 months of crizotinib in a young patient with pulmonary IMT. Therefore, ceritinib was administered as a second-line treatment leading to disease control for three months.

From a future perspective, another question is at what point should ALK inhibitors be introduced? Given the aggressiveness of EIMS, is there any point in introducing adjuvant therapy after complete surgery, and if so, for how long? Furthermore, the use of ALK inhibitors in the neoadjuvant setting could be an excellent opportunity however, evaluating the benefit of adjuvant therapy is extremely challenging in such a rare disease. In the real world, patients tend to perform worse than clinical trial participants, limiting the ability to initiate and complete adjuvant therapy due to postoperative complications and decreased treatment tolerance. In addition, the neoadjuvant approach offers the unique opportunity to study radiological and adaptive responses of tumors to systemic therapy, which can be potentially used for prognostic purposes and to tailor adjuvant treatment strategies. At least one case of neo-adjuvant use of an ALK inhibitor has been described. In 2015, Rafee et al. reported a case in which an excellent response after 8 months of treatment with ALK inhibitor allowed for subsequent surgical management [66]. This also opens up many questions regarding the best way to treat our patients.

## 7. Conclusions

IMT is a challenging disease and a perfect example of personalized medicine with the wide use of NGS. The best TKI strategy for ALK-positive IMTs is still yet to be determined. In addition to targeting ALK, more data are needed regarding the effectiveness of other TKIs, as well as regarding alternative treatment modalities such as chemotherapy and immunotherapy. Recently, long-term remission with rituximab for up to 19 months has been reported in a young, heavily pre-treated patient with lung IMT [86].

The accumulation of further cases of IMTs is crucial. These data will not only allow for a better understanding of this rare tumor type but will also suggest rational targeted therapeutic strategies for existing TKIs, based on the genomic profile of the tumor.

Considering the rarity of IMS, it is unlikely that we will get an answer to these questions soon, thus the inclusion of patients in clinical studies should always be encouraged.

## Figures and Tables

**Figure 1 cancers-14-03662-f001:**
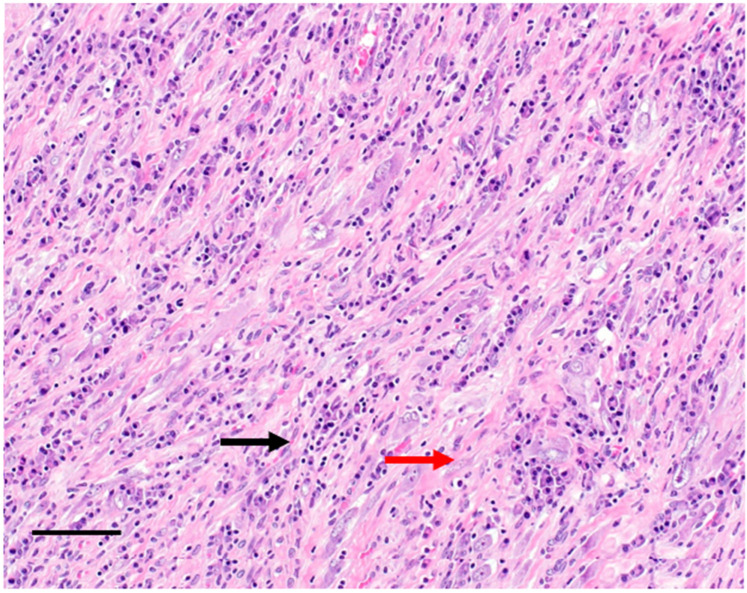
Hematoxylin eosin stain of an inflammatory myofibroblastic tumour composed of myofibroblast cells (red arrow) and inflammatory infiltrates (black arrow). Bar = 20 µm.

**Figure 2 cancers-14-03662-f002:**
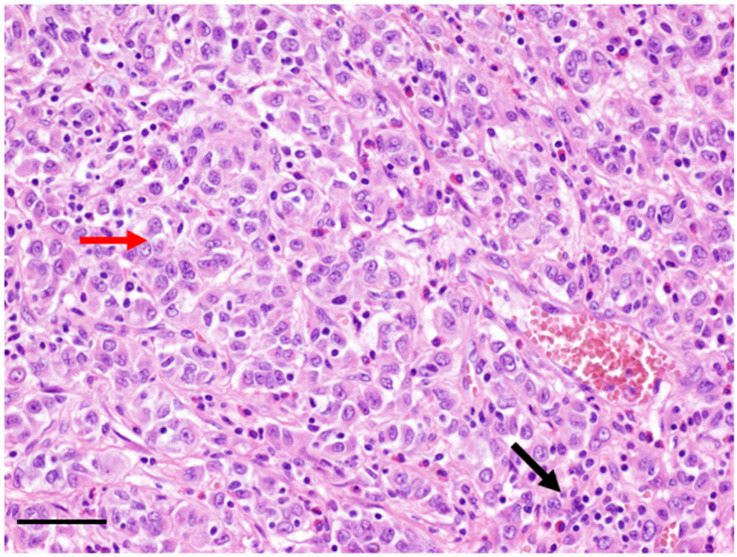
Hematoxylin and eosin stain of an EIMS composed of epithelioid cells (red arrow) and inflammatory infiltrates (black arrow). Bar = 10 µm.

**Figure 3 cancers-14-03662-f003:**
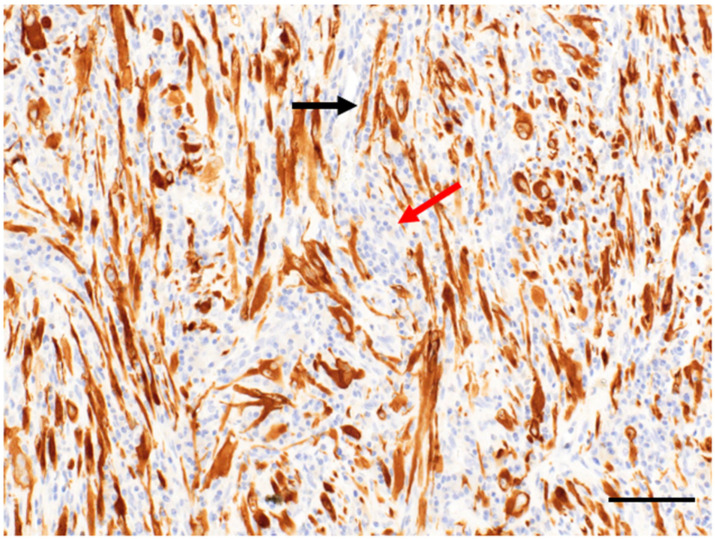
Immunostaining of an IMT with an antibody against anaplastic lymphoma kinase (ALK) and a peroxidase conjugated secondary antibody. Positive cells were stained with diaminobenzidine (black arrow). Nuclei were counterstained with hematoxylin (red arrow). Bar = 10 µm.

**Figure 4 cancers-14-03662-f004:**
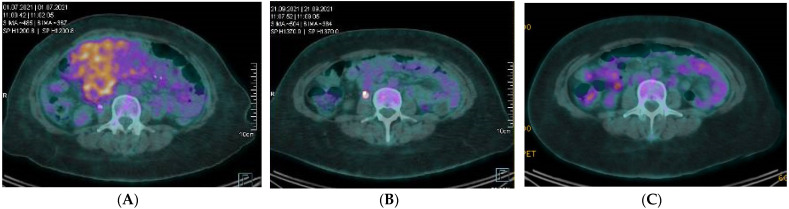
Positron emission tomography (PET) scans showing evolution in time of the abdominal EIMS with large hypermetabolic mesenteric mass, associated with several hypermetabolic intraperitoneal nodules (**A**) A clear morpho-metabolic regression of the mesenteric mass and secondary intraperitoneal implants previously visualized 3 and 6 months after targeted therapy with alectinib (**B**,**C**). (**A**) Baseline. (**B**) After 3 months of alectinib. (**C**) After 6 months of alectinib.

**Table 1 cancers-14-03662-t001:** Clinical and pathological features of the reported cases of epithelioid inflammatory myofibroblastic sarcoma.

Author, Year	Number of Cases	Age Range	M:F	Location	Treatment	Recurrence/ Metastasis	Dead of Disease	ALK IHC Pattern	Fusion Partner
Butrynski J et al., 2010 [32]	1	44	M	Intraabdominal	SE, HPP, CT Imatinib ALKi (Crizotinib)	Yes	-	Nuclear membrane	*RANBP2–ALK*
Mariño-Enríquez A et al., 2011 [17]	11	6–63	10:1	Intraabdominal	SE (2) SE + CT (5) SE +CT+RT (2) SE + CT +ALKi (1) (Experimental ALK inhibitor) NA (1)	10	5 out of 8 in which follow-up available	Nuclear membrane 9 out of 11 Cytoplasmic with perinuclear attenuation in 2 of 11	9 *ALK* translocation *RANBP2–ALK* in 3 of the 9
Li J et al., 2013 [65]	2	19 and 39	1:1	Pelvic cavity	SE SE + CT	2	1	Nuclear membrane	*RANBP2–ALK*
Kozu et al., 2014 [34]	1	57	M	Pleural cavity	CT + ALKi (ALK inhibitor no precision)	Yes	SE	Cytoplasmic pattern with perinuclear accentuation	*RANBP2–ALK*
Kimbara S et al., 2014 [35]	1	22	M	Intraabdominal	SE + CT + ALKi (Crizotinib)	Yes	No F/U after 10 months	Nuclear membrane	*RANBP2-ALK*
Kurihara-Hosokawa K et al., 2014 [36]	1	22	M	Intraabdominal	SE + ALKi (Crizotinib)	Yes	No	Nuclear membrane	*RANBP2-ALK*
Rafee S et al., 2015 [66]	1	55	F	Intraabdominal	CT ALKi (Crizotinib) SE	Yes		Nuclear membranous staining	Only *ALK* FISH done, fusion parnters unknown
Fu X et al., 2015 [67]	1	21	M	Lung	SE ALKi (Crizotinib)	Bone metastasis	No F/U after 3 months	Cytoplasmic	Only *ALK* FISH done, fusion parnters unknown
Bai Y et al. 2015 [68]	1	65	M	Intraabdominal		Yes	No F/U	No mentioned	No mentioned
Wu H et al., 2015 [69]	1	47	F	Intraabdominal		Yes	Yes	Nuclear membrane	*RANBP2-ALK*
Sarmiento et al., 2015 [37]	1	71	M	Pleural	SE + ALKi (Crizotinib, switch for 2nd line)	No	Alive	ALK positive—pattern not mentioned	Only *ALK* FISH done—fusion partners unknown
Lee JC et al., 2015 [70]	5	16–76	3:2	Liver (1) Lung (1) Intraabdominal (3)	SE (3) SE + CT (1) SE + CT + RTH + ALKi (1) (Crizotinib)		4 DOD within 12 months 1 Alive at 33 months	Cytoplasmic (2) Nuclear (3)	*RANBP2-ALK* fusion in the 2 other cases
Liu Q et al., 2015 [39]	1	22	M	Intraabdominal	SE + ALKi (Crizotinib)	Yes	Alive	Nuclear membrane	*RANBP2-ALK*
Yu L et al., 2016 [19]	5	15–58	2:3	Intraabdominal	37: SE 55: SE, SE, CT 22: SE, recurrence, ALKi (Crizotinib) 58: SE, CT 15: SE	Yes 3 out of 5	37: No recurrene, alive 55:	Nuclear membrane pattern in 4 cases Cytoplasmic staining with perinuclear accentuation fashion in 1case	5 tumors showed *ALK* gene rearrangement
Jiang et al., 2017 [71]	1	45	M	Intraabdominal	SE + ALKIi adjuvant (Crizotinib –stop for severe vomiting + elevation AST et ALT) ALki for mestatatic disease (Crizotinib) –tumor lyse syndrome, DOD	Metastasis to liver, spleen, small intestine et al.	Yes	Cytoplasmic	*EML4-ALK*
Lee et al., 2017 [70]	9	7 months-76	6:3	Intraabdominal (*n* = 7) Lung (*n* = 1) Liver (*n* = 1)	SE (9) 2 treated with ALKi (Crizotinib)	Yes 9 out of 9	6	4 Nuclear membrane (*n* = 4) and 5 cytoplasmic staining (*n* = 5; 4 with perinuclear accentuation).	*RANBP2-ALK (3)* *RRBP1-ALK (5)* *No mentionned (1)*
Fang et al., 2017 [72]	1	52	F	Small bowel	?	?	Yes (8 months)	?	*ALK done, fusion partners unknown*
Du X et al., 2018 [73]	1	26	M	Intraabdominal	SE + CT	Yes	Yes	Cell nuclei	*RANBP2-ALK*
Xu X et al., 2019 [74]	1	28	M	Intraabdominal	ALKi (Crizotinib then Brigatinib)	Yes	No		*RANBP2-ALK*
Hallin M et al., 2018 [75]	1	-	-	-	-	-	-	-	*ALK*-unknown
Xu P et al., 2019 [33]	1	35	F	Gastric	SE	No	No (limited F/U)	Cytoplasmic	*N*/*A*
Zhang S et al., 2019 [62]	1	46	F	Intraabdominal	SE ALKi at progression (Crizotinib), no response and then multi-targeting tyrosine kinase inhibitor (Anlotinib)	Yes	Yes (16 months)	Yes	2p23 *ALK* gene rearrangement
Liu D et al., 2020 [76]	1	N/A	N/A	Sigmoid colon	SE + ALKi (Crizotinib)	Yes	N/A	Perinuclear	*RANBP2–ALK*
Kopelevich A et al., 2020 [77]	1	17	M	Renal	SE + ALKi (Crizotinib, and then Alectinib)	Yes	N/A	N/A	*RANBP2–ALK*
Zilla et al., 2021 [78]	1	80	M	Right groin	SE	?	No G/U	Nuclear membrane	*RANBP2-ALK*
Chopra S et al., 2021 [79]	1	72	F	Brain	SE At progression ALKi (Alectinib)	Yes	At 4 months	Cytoplasmic	*VCL-ALK*
Gadeyne L et al., 2021 [80]	1	27	F	Cutaneous	SE	No	No	Very clear cytoplasmic staining with perinuclear accentuation	*RANBP2-ALK*
Collins K et al., 2022 [81]	1	43	F	Uterus	SE + CT	Yes	No	Nuclear membrane	*RANBP2-ALK*
Wang S et al., 2022 [45]	1	42	F	Intraabdominal	SE Crizotinib At progression (PD after 5 months) Alectinib At progression (PD after 5 months) Ceritinib At progression (PD after 6 months) Lorlatnib (SD after 5 months)	Yes	No		*PRRC2B-ALK*
Current case	1	39	F	Intraabdominal	ALKi (Alectinib)	Yes		Nuclear membrane	*RANBP2-ALK*

N/A—not available. SE—surgical excision. CT—chemotherapy. RT—radiotherapy. ALKi—ALK inhibitor. F/U—Follow up.

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
