# Peer review of "Inflammatory Myofibroblastic Tumour: State of the Art"

_cancers, 2022, doi:10.3390/cancers14153662_

Round 1

Reviewer 1 Report

This review article provides a detailed information on the use of therapies for inflammatory myofibroblastic tumor (IMT).

Overall, this is a clear and concise, and well-written manuscript. The introduction is relevant and appropriate.

Sufficient information about the previous study findings is presented for readers to follow the manuscript.

Minor suggestions:

1) Line# 33: The author mentions IMT as an ultra-rare entity. It is recommended to provide some statistics with some information on incidence and prevalence for IMT.

2) In this manuscript, there is excessive use of abbreviations and short-forms. It is highly recommended to provide the full-forms or definitions of the abbreviations for a broad range of readers to understand the terminologies.

3) Within this manuscript, there are couple of typos with double spacebars (Line# 52, 249, and few more).

4) For the immunohistochemistry (IHC) images, it is recommended to please provide a scalebar, some information on the type of staining if it is an H&E staining, what color represents positive staining and contour staining (for nucleus).

Author Response

Dear reviewer,

Thank you very much for reviewing our manuscript.  We addressed all the queries and followed the recommendations suggested

1) Line# 33: The author mentions IMT as an ultra-rare entity. It is recommended to provide some statistics with some information on incidence and prevalence for IMT.

Now on Line #50-52 we added the information on the incidence and prevalence of IMT.  “ The scarce existing data show that it occurs most often in children and young individuals with a prevalence ranging from 0.04% to 0.7% regardless of gender and race in the world population”

2) In this manuscript, there is excessive use of abbreviations and short-forms. It is highly recommended to provide the full-forms or definitions of the abbreviations for a broad range of readers to understand the terminologies.

We have spelled out all the abbreviations used in the manuscript, the first time they appear in the text.

3) Within this manuscript, there are couple of typos with double spacebars (Line# 52, 249, and few more).

We corrected all the typos (Line #52, 249 and others we identified)

4) For the immunohistochemistry (IHC) images, it is recommended to please provide a scalebar, some information on the type of staining if it is an H&E staining, what color represents positive staining and contour staining (for nucleus).

We added a spacebar to the IHC images and described in the legend the type of staining and counterstaining that has been used. 

Reviewer 2 Report

Inflammatory myofibroblastic tumour (IMT) is a very rare type of tumor. The pathogenesis, diagnosis and treatment are all understudied. In this article, the authors made an effort to review what is currently known about this disease and summarized most of the cases reported in the literature. This article may give readers a comprehensive up-to-date view of this disease, while the text shall be substantially revised to improve it readability. There are numerous syntax and grammatical errors and misspellings throughout the manuscript text, including a few mentioned below.

1.       Please reword the sentence in the abstract “It is in this context that data on targeted therapies are most important.”

2.       The last sentence of the Abstract states “The aim of this article is to review the latest published data on the use of systematic therapy and to publish an additional case of epithelioid inflammatory myofibroblastic tumor with RANPB2-ALK fusion with long response to a tyrosine kinase inhibitor.” However, there is not new case reported in this manuscript. Please clarify which is the additional case.

3.       Please reorganize the sentence that runs from Line 140 to Line 146.

4.       Change “greater” in Line 169 to “older”.

5.       In Lines 180-181, “In the ALK-positive group patient two out of six patients…”, the “patient” is extra.

6.       In Line 233, “3 came were positive for ROS1” should be “3 cases…”

7.       Line 281, what is “EIMT”?

8.       The case described in Lines 341-343 is the same case in Lines 335-340? If yes please combine these two paragraphs.

9.       There are three images included in the manuscript, but they are not mentioned in the text.

Author Response

Dear reviewer,

Thank you very much for reviewing our manuscript.  We addressed all the queries and followed the recommendations suggested

  1. Please reword the sentence in the abstract “It is in this context that data on targeted therapies are most important.”

The sentence in the abstract has been reworded as follows:  “ In this context, data on targeted therapies are most important”

  1. The last sentence of the Abstract states “The aim of this article is to review the latest published data on the use of systematic therapy and to publish an additional case of epithelioid inflammatory myofibroblastic tumor with RANPB2-ALK fusion with long response to a tyrosine kinase inhibitor.” However, there is not new case reported in this manuscript. Please clarify which is the additional case.

The new case is presented in lines 338-341, in the table 1 (last line) and the new figure 4 in the manuscript

  1. Please reorganize the sentence that runs from Line 140 to Line 146. The sentences on page 140-146 are “The authors therefore conclude that glucocorticoids are especially recommended as a basic part of an integrated therapy for ENT IMT. Another case report of a 48-year-old patient with inoperable maxillary sinus IMT with complete response after concomitant radiotherapy and prednisone was published, suggesting that radiotherapy with steroids could be an effective alternative to surgery, especially in the treatment of maxillary IMT (28). »

The sentence has been reworded as follows:  “ The authors, therefore, conclude that glucocorticoids are especially recommended as a basic part of integrated therapy for ear, nose, and throat IMT. Another case report of a 48-year-old patient with inoperable maxillary sinus IMT with complete response after concomitant radiotherapy and prednisone have been published. In this context, combination of radiotherapy with steroids appears to be an effective alternative in non-operable maxillary IMT”

  1. Change “greater” in Line 169 to “older”. The reviewer refers to the world on line 179. It has been done.

Thank you again for reviewing our manuscript

Round 2

Reviewer 2 Report

The authors appropriately revised the manuscript in response to this reviewer.